# Optimal Nutrition Parameters for Neonates and Infants with Congenital Heart Disease

**DOI:** 10.3390/nu14081671

**Published:** 2022-04-17

**Authors:** Alina-Costina Luca, Ingrith Crenguța Miron, Dana Elena Mîndru, Alexandrina Ștefania Curpăn, Ramona Cătălina Stan, Elena Țarcă, Florin-Alexandru Luca, Alexandra Ioana Pădureț

**Affiliations:** 1Department of Pediatrics, Faculty of Medicine, Gr. T. Popa’ University of Medicine and Pharmacy, 700115 Iasi, Romania; acluca@yahoo.com (A.-C.L.); lucmir@gmail.com (I.C.M.); eledanamindru@gmail.com (D.E.M.); 2Department of Biology, Faculty of Biology, “Alexandru Ioan Cuza” University of Iasi, Bd. Carol I, 20A, 700505 Iasi, Romania; 3Sfânta Maria’ Emergency Children’s Hospital, 700309 Iasi, Romania; ramonacatalina.stan29@yahoo.com (R.C.S.); paduret.alexandra@gmail.com (A.I.P.); 4Department of Surgery II – Pediatric Surgery, ”Grigore T. Popa” University of Medicine and Pharmacy, 700115 Iasi, Romania; elatarca@gmail.com; 5Department BMTM, “Gheorghe Asachi” Technical University of Iasi, 700050 Iasi, Romania; florin.alexandru.luca@gmail.com

**Keywords:** congenital heart disease, nutrition, enteral feeding, parenteral feeding

## Abstract

Congenital heart defects are known causes of malnutrition. Optimal nutritional management is paramount in improving short and long-term prognosis for neonates and infants with congenital heart malformations, as current strategies target preoperative and postoperative feeding requirements. Standardized enteral and/or parenteral feeding protocols, depending on the systemic implications of the cardiac defect, include the following common practices: diagnosing and managing feeding intolerance, choosing the right formula, and implementing a monitoring protocol. The latest guidelines from the American Society for Parenteral and Enteral Nutrition and the European Society of Paediatric and Neonatal Intensive Care, as well as a significant number of recent scientific studies, offer precious indications for establishing the best feeding parameters for neonates and infants with heart defects.

## 1. Introduction

Congenital heart malformations (CHD) are structural and functional abnormalities that occur during embryogenesis. CHDs are classified based on morphological and pathophysiological criteria [1]:Cardiac anomalies with increased pulmonary flow such as septal defects with left to right shunt and no pulmonary obstruction;Cardiac defects with decreased pulmonary flow;Congenital anomalies that spare the septum;Severe cardiac anomalies;Congenital heart disease which is asymptomatic until adulthood.

Various animal models have been used to reproduce the pathological phenotypes encountered in humans by disrupting the molecular pathways involved in the myocardial specification and differentiation, or subsequent stages of morphogenesis. In an attempt to establish the genetic basis of these conditions, research in recent decades has focused on the genomic analysis of familial cases of CHD and gene sequencing in cohort studies.

The efforts to improve the management of these conditions have led to metabolomics studies focused on the identification of the metabolic changes that occur in heart cells subjected to chronic hypoxia by describing the differences in metabolic profiles between cyanotic and acyanotic congenital heart malformations [2].

This review highlights the specific metabolic changes in patients with CHD and summarizes studies regarding appropriate enteral and parenteral nutrition.

### Malnutrition Determinants in Patients with Congenital Heart Defects

The evaluation of nutritional status and growth in children is based on WAZ (weight for age), WHZ (weight for height), and HAZ (height for age) parameters. A cut-off Z-score < −2 classifies malnutrition into underweight (low WAZ), stunting (low HAZ), and wasting (low WHZ) [3].

Children with multiple anthropometric deficits have a heightened risk of mortality, and more so those with congenital heart anomalies. Stunting is associated with cyanotic heart diseases and pulmonary hypertension association (PHA), while acyanotic heart anomalies are usually accompanied by wasting [4]. The appearance and progression of malnutrition in these patients is dictated by the hemodynamic impact of heart lesions, the occurrence of heart failure, delayed surgical repair, prolonged intubation, and feeding intolerance.

Pulmonary hypertension (PHA) has the strongest association with pre-operative malnutrition [5]. The underlying mechanisms of PHA are mainly related to hypoxia-induced adaptive responses and increased pulmonary flow. In CHD with left to right shunt, there is excessive pulmonary blood flow, which leads to endothelial dysfunction, vascular remodeling, and progressively higher pulmonary vascular resistance (PVR). Hypoxic pulmonary vasoconstriction is the physiological response to a regional decrease in oxygen availability and it is crucial for matching ventilation and perfusion, albeit whilst causing an increase in PVR [6]. Once installed, PHA causes right ventricular failure, which in turn leads to gastrointestinal edema, malabsorption, altered microbiome, and fluid retention [7]. The medication used to prevent fluid retention in this scenario, such as diuretics, causes vitamin B1 deficiency, a micronutrient involved in carbohydrate and branched-chain amino acid metabolism [8].

As a response to hypoxic conditions, Na^+^/K^+^-ATPase activity is limited in order to prevent unnecessary ATP expenditure [9]. Studies showed that Na^+^ pump inhibition is related to malabsorption of nutrients and electrolytes [10].

Hypoxia also induces antioxidants depletion and modifies the expression of the genes-coding enzymes involved in glucose metabolism. HIF-1 (hypoxia induced factor) promotes lactate dehydrogenase A, which turns pyruvate into lactic acid in the final staged of glycolysis [11]. In chronic hypoxia, glucose and lipid storage are depleted [7].

Another possible theory that has been investigated is the involvement and correlation of mitochondria with the negative outcomes in patients with CHD, alongside the imbalance between biogenesis, fission, fusion, and mitophagy, which were indicated as the main mechanisms behind mitochondrial dysfunction [12]. Mitophagy can be triggered by exogenous substances, and their use in increasing cardio protection is still under investigation [13,14].

Once diagnosed, in the case of a positive prognosis, patients with CHD either immediately undergo a surgical repair or are subjected to pharmacological therapy until the surgical intervention is possible, which makes building a trust relationship between parents and doctors of utmost importance [15]. Meeting the perioperative needs of children scheduled for a cardiopulmonary bypass has an important impact on the postoperative clinical outcome.

Similarly, patients reaching puberty before surgical correction face the risk of heart failure, because of the metabolic changes that normally occur in the heart tissue. The glucose-dominant metabolism active during prepuberty switches to fatty-acid dominant metabolism during puberty. This switch is impaired in hearts affected by chronic hypoxia, as is the case with a congenital heart malformation. In this scenario, the outcome of these patients could be improved by pioglitazone administration during puberty [14,16].

Enteral and parenteral feeding have standardized parameters in patients with CHD before and after surgery. Minor cardiac anomalies with an insignificant hemodynamic and systemic impact fall under the general principles of feeding, while complex cardiac anomalies require personalized regimens.

## 2. General Principles for Enteral Feeding in Neonates and Infants with Congenital Heart Disease

During the neonatal period, enteral feeding (EN) should be initiated as soon as possible, with breast milk or formulas that ensure the energy and macronutrients requirements are met. A term ill newborn initially needs 40–60 kcal/kg/day, with a gradual increase to 90–120 kcal/kg/day. The daily requirement of carbohydrates is between 9–14 g/kg/day (40–50% of total calories), proteins—1.8–2.2 g/kg/day (7–16% of total calories), and lipids—of 4–6 g/kg/day (34–35% of total calories). In neonates with a CHD, the daily caloric intake should be up to 50% higher, but without exceeding 150 mL/kg/day liquid volume, and daily protein requirements can be as high as 3 g/kg/day.

Human milk as well as formulas available on the market have measurable levels of calories, lipid, glucose, and protein concentrations.

The evolution of newborns with heart abnormalities associated with patent ductus arteriosus (PDA) is burdened by a high risk of necrotizing enterocolitis (NEC), due to decreased intestinal blood flow, especially during diastole [17]. Secondary measures of the prophylaxis of NEC in this category of patients are based on the initiation of parenteral nutrition, in a sufficient volume to ensure the support of the maturation process of the intestinal mucosa, without a high risk of energy waste. It is considered that 10–20 mL/kg/day is a sufficient amount for enteral administration [18]. Concentrated feeding formula, with fortified human milk or a high calorie formula, is a way to deliver more calories and nutrients when fluid intake volume is limited [19]. Breast milk is considered the ideal source of nutrition for all infants, providing important quantities of immunoglobulin A, proteins, and amino acids, and by doing so, lowers the risk of respiratory and autoimmune diseases [20]. Human milk has also been proved to reduce the chances of NEC and other gastrointestinal malfunctions [20,21]; provided that the mother meets optimal daily nutritional values and is neither malnourished nor overweight, the alimentation should be enriched with vitamins [22].

Research shows that breastfeeding can be an option for CHD infants, although there are persistent concerns regarding the energy expenditure, which traditionally was considered to be greater than bottle feeding [18,23]. The frequency and duration of feeding as well as the preparation method must be assessed to avoid complications that may arise from incorrect preparation. Both over-dilution and excessive concentration can result in poor growth, electrolyte imbalance, and gastrointestinal impairment [24].

Feeding and crying are activities that require energy consumption. A feeding session is the only activity that can be controlled in terms of running time; the current recommendation is to limit feeding to a maximum of 30 min/feeding episode [25].

In preterm babies, gavage feeding, via nasal or oral tubes, using intermittent bolus feeding (iBF), or continuous feeding (CF) techniques, is the best option to ensure enteral administration of food. The widespread infusion methods for preterm neonates are established [26]:(1)iBF (10–20 min infusion every 2–3 h),(2)Slow infusion intermittent feeding (30–120 min infusion every 2–3 h),(3)CF (continuous infusion over 24 h),(4)Semicontinuous feeding (feeding every 15 min throughout the day with one-fourth of the hourly feed volume).

In choosing the appropriate infusion type, one must acknowledge the maturity of enteric neuroregulation. The migrating motor complex in term infants follows the classical three phased pattern [27], whereas in preterm infants, phase three, which is supposed to ensure the passage of the intestinal bolus, has low-amplitude, non-propagating pressure waves [26]. The slow infusion of food in preterm infants results in a postprandial response similar to that of term infants, and helps with the maturation of the neuromuscular function. The characteristics of fasting motor activity mature within 10 days to resemble that of term infants if enteral feeding is commenced, irrespective of the gestational age (GA) at birth [26].

The hemodynamic aspects of splanchnic circulation are also critical in choosing the best feeding method. CF ensures a constant blood flow in the mesenteric circulation, but over time, the circulation via the superior mesenteric artery (SMA) becomes unable to adapt to the higher demands that come with higher feeding volumes or concentration [28]. iBF ensures a postprandial response similar to adults and is therefore more physiological.

Gut endocrine function and the balance between anabolism/catabolism differs in dependence of the feeding regime chosen. Insulin and amino acids concentration increases in feeding-induced cycles, with CF ensuring a lower stimulation than iBF. Muscle protein synthesis is two times lower in CF compared to iBF [29]. Continuous protein supplementation instead inhibits protein synthesis and induces AAs catabolism. If, however, CF is required, especially in neonates and infants who are scheduled for a blood transfusion, offering a supplement of leucine can boost the AAs synthesis. CF is more energy efficient, and diminishes behavioral stress, albeit whilst inhibiting protein gain and negatively interfering with gut paracrine function. Therefore, iBF is the better option, and slow infusion intermittent feeding with fasting intervals encourages better gastric emptying and paracrine function, while leading to calcium and lipids loss, more so than with gravity feeding [26,29,30].

In gavage-fed newborns, monitoring the gastric residue before each feeding is not necessary [31]. Some studies suggest that a gastric residue above 25% of the volume of the previously administered meal determines the need to deduct an equal volume from the amount of milk to be administered. However, ESPNIC challenges the inaccurate measurement of GRV and the level of acceptability for the gastric residue, and advises against the routine measurement of GRV [32].

Gavage feeding may also be initiated in ill term neonates that cannot be breast or bottle fed but have a functional digestive system.

Monitoring the effectiveness of enteral feeding involves daily weight gain evaluation, aspect and frequency of stools, diuresis, and weekly measurements of height and head circumference.

## 3. General Principles of Parenteral Feeding in Neonates and Infants with Congenital Heart Disease

Parenteral feeding meets the nutritional needs by administering nutrients via the venous pathways. Parenteral nutrition can be total or partial, depending on the newborn’s ability to tolerate enteral feeding. The venous routes can be peripheral or central, and choosing them means taking into account the complications that may occur, depending on the type of venous approach.

The IV fluid rate for hospitalized children follows the 4:2:1 rule—4 mL/kg/h for the first 10 kg of weight, 2 mL/kg/h for the next 10 kg, and 1 mL/kg/h for every kg past 20 kg. Fluid intake in term infants is 60–70 mL/kg on day 1 and increases to 100–120 mL/kg by day 2 or 3. Premature infants may receive 70–80 mL/kg on day 1 and slowly advance to 150 mL/kg/day [33].

Choosing the right IV fluid is paramount. The neonate renal function is unable to balance sodium secretion/excretion. Therefore, IV fluids must contain sodium. Considering the normal range of plasma osmolality as 275–290 mOsm/kg, normal saline solution (0.9% NaCl) is slightly hypertonic and Ringer’s lactate solution is isotonic and hyponatremic. Hypotonic solutions should not be used, due to their risk of inducing hyponatremia. According to The European Society for Paediatric Gastroenterology Hepatology and Nutrition (ESPGHAN), electrolytes can be administered in infants under 5 kg from day 1, depending on the blood levels (see Table 1 for dosage recommendations regarding electrolytes). When calculating the total caloric intake, macro and micronutrients kcal values per gram should be taken into account. A total of 1 g of protein provides 4 kcal and the amount of protein should provide 10–15% of the daily calories. A total of 1 g of lipids provides 9 kcal and lipids should account for about 30–35% of the total daily calories. A total of 1 g of glucose provides 4 kcal and the amount of carbohydrates should provide 60–65% of the daily calories [33].

The total fluids and nutrients for the ill newborns who require total or partial parenteral nutrition are calculated using birth weight in the first 3 days of life, then daily weight. The amount of fluid received by enteral feeding or other administered fluids (e.g., fluids used to dilute medication, or blood products) should be deducted from the total fluid calculated for 24 h.

The fluid intake should be increased by 10–20 mL/kgc/day and should reach a maximum of 150 mL/kgc/day during the first week, provided that no significant fluid losses occur.

The infusion rate of the glucose solution should initially be 2.5–5 mg/kgc/min and it can be increased daily by 1–2 mg/kgc/min to a maximum of 12 mg/kgc/min, while monitoring blood glucose levels. Peripheral vein access is suitable for glucose solutions of 12.5%, whereas a central vein access allows administration of 25% glucose solutions.

The minimum amount of protein to avoid a negative nitrogen balance is 1.5 g/kg/day, and the maximum intake of amino acids should not exceed 3 g/kg/day. The amount of amino acids should be reduced to 1.5 g/kg/day in situations requiring fluid restriction, renal impairment, acidosis, hyperammonemia, and hepatic impairment associated with prolonged parenteral nutrition [34].

The nitrogen ratio, calculated according to the formula (RCN = non-protein calories (glucose + lipids)/0.16 × protein (g)), ensures adequate caloric intake for the amount of protein administered. Values above 250 indicate that protein intake may be increased as needed. Values below 150 indicate the need to increase non-protein calories or decrease protein intake (13).

Lipid administration should be initiated from day 1–2 of life, starting with 0.5–1 g/kgc/day, and then increasing the amount administered by 0.5 g/kgc/day to a maximum of 3 g/kgc/day. Lipid emulsions are calculated from the total volume of liquids. The lipid solution is infused undiluted for 18–24 h at a rate of 0.5–1.5 mL/h. The lipid emulsion is administered on a separate venous line or in a triple route with a brown connection [35].

The syringes and tubing used for parenteral administration of lipids must be protected from light, because under the action of light and especially phototherapy, lipids can peroxide, generating toxic compounds that can damage tissues.

Parenteral administration of lipids is contraindicated in patients with severe sepsis, severe pulmonary disease, high pulmonary vascular resistance, and high bilirubinemia; this indicates the necessity for exsanguination transfusion [36].

All of the above mentioned enteral and parenteral optimal nutrients for newborns and neonates, as well as the special considerations, can be visualized in Table 2.

## 4. Special Considerations in Patients with Congenital Heart Disease

Patients with CHD often exhibit failure to thrive and poor nutrition. Hypermetabolic state, swallowing difficulties, upper respiratory tract infections, gastroesophageal reflux (GERD), malabsorption, and genetic syndromes are important etiological factors [37]. The hormonal imbalance caused by significant stress together with a metabolic shift toward fatty-acid oxidation and poor carbohydrate use lead to the impaired use of nutritional resources [38].

The European Society of Paediatric and Neonatal Intensive Care opines in favor of starting enteral nutrition within 24 h from admission [18], provided that gastrointestinal anomalies, vomiting, diarrhea, NEC, or lactic acidosis are not present. Furthermore, introducing EN in neonates and infants on mechanical and/or pharmaceutical hemodynamic support was associated with lower mortality, because research showed that that EN improves gut paracrine function, and does not alter the intestinal barrier or increase the risk of sepsis. These results apply also to older children. For neonates and children who are stable on vasoactive drugs and/or after cardiac surgery, early EN is also recommended. There is still much debate on whether trophic enteral nutrition (TF) is useful in preventing intestinal tract complications, especially after hypoplastic left heart syndrome (HLHS) corrective procedures. Recent studies showed that TF may improve the clinical outcome of these patients by shortening mechanical ventilation periods and allowing for earlier enteral feeding. Patients with ductal dependent-CHD develop systemic desaturation and decreased abdominal blood flow, associated with hypoperfusion during diastole, all of which lead to mesenteric ischemia and a theoretical higher risk for NEC. However, Becker et al. argue that, according to their cohort study results, NEC is not significantly associated with preoperative enteral feeding in patients with ductal-dependent CHD, naming only single-ventricle heart defects, mainly HLHS, as a factor linked to the higher percentage of NEC. Mortality rates were higher in patients with NEC, and amongst them, 7 out of 10 were premature neonates. Neonates should receive EN while being monitored for systemic and gut perfusion abnormalities [39].

Intraoperative fluid requirements in children, depending on the type of surgery, range between 1 mL/kg/h and 15 mL/kg/h, and premature babies may receive up to 50 mL/kg/h. A general rule states that for every ml of blood lost during surgery, 1.5 mL of isotonic crystalloid solution should be infused [40].

During the first 12 h after surgery, fluid IV intake falls under the 2:1:0.5 rule, which is 2 mL/kg/h for children weighing up to 10 kg, 1 mL/kg/h for the next 10 kg, and 0.5 kg/h for every kg exceeding 20, using isotonic fluids [41]. If enteral nutrition cannot be initiated after 12 h postoperatively, then hypertonic fluids should be administered according to the 4:2:1 rule mentioned in Section 2.

Regarding specific nutritional parameters, ESPNIC recommends that energy intake in ill patients during the acute phase should not exceed the resting energy expenditure. The resting energy expenditure (REE) in these patients is significantly increased and negatively impacts cardiac output and inflammatory responses. In ICUs, indirect calorimetry can be used to assess the energy needs. The formulas used are: REE (kcal/day) = ((VO_2_ × 3.94) + (VCO_2_ × 1.11)) × 1440 min/day, and the respiratory quotient (RQ) = VCO_2_/VO_2_; the RQ is normally within the range of 0.67–1.3. Eligible for applying these measurements are patients with weight < 5 percentile or > 85 percentile for age, those with >10% variation in weight during ICU stay, or patients who cannot be weaned from respiratory support [42]. If an indirect calorimetry cannot be used, then the Schofield equation for age and gender, using weight, which has been proved to be the least inaccurate in determining REE, is therefore a good substitute [43]. After the acute phase, energy intake should be based on REE, physical activity, rehabilitation, and growth.

### 4.1. Glucose Intake

The balance between hyper and hypoglycemia in critically ill patients with CHD is difficult to maintain. The risk for hypoglycemia is higher in neonates, children with endocrinopathies, and those over-fed, as well as in patients who undergo a certain period of fasting before general anesthesia. Mild hypoglycemia combined with hypoxia and/or ischemia determines severe neurological impairment, measurable during up to 18 months of follow-up, and increases the chances of a negative outcome for patients in the Pediatric Intensive Care Unit. Hyperglycemia, on the other hand, resulting from an impaired glucose metabolism, leads to cellular death, electrolyte imbalance, and neurological impairment. In the acute phase, endogenous glucose production and a certain level of insulin resistance cover most of the glucose requirements, tilting the scales towards hyperglycemia. During the recovery phase, an equilibrium installs, allowing for more glucose to be administered. A parenteral glucose intake of 2.5 mg/kg/min (3.6 g/kg/day) in the acute phase and 5.0 mg/kg/min (7.2 g/kg/day) in the recovery phase is recommended, with 10% glucose solutions being preferred [44].

### 4.2. Protein Requirements

A protein intake which avoids a negative protein balance is standard for healthy patients. However, those with CHD exhibit a higher degree of protein breakdown, concomitant with a higher positive protein balance achieved by up to 3.1 g/kg/day of protein intake. During a direct dependence on protein intake, endogenous glucose and lipolysis levels increased in these patients. Administering 1.5 g/kg/day for infants and 0.8 g/kg/day in children seems to be the best approach for these cases. Higher levels do not produce a positive outcome, mostly due to anabolic resistance, according to ESPNIC.

The American Society for Parenteral and Enteral Nutrition’s guidelines offer a different view. Suggestions for protein supplementations in critical CHDs are: 2–3 g/kg/day for ages 0–2 years, 1.5–2 for ages 2–13 years, and 1.5 for those between 13 and 18 years of age [24,45].

### 4.3. Lipid Intake

Lipid intake should not exceed 3 g/kg/day and doses should be modified by monitoring triglycerides levels. Composite lipids emulsions could be a better option than other available solutions, particularly pure soy oil lipid solutions, due to their antioxidative effects, and their ability to improve cholestasis and liver dysfunction. Intralipid, a 20% IV fat emulsion, along with parenteral nutrition, allows for reaching the daily caloric requirement, while maintaining an adequate osmotic load. A total of 0.5 mg/kg/day intralipid is enough to prevent lipid deficiency [46].

### 4.4. Formulas

Most children with CHD have a diminished feeding capacity and require fluid restriction. A formula rich in protein and energy should be used to provide larger amounts of nutrients, while balancing the osmotic load carefully to prevent osmotic diarrhea. In that regard, the osmotic load of formulas should not exceed 450 mOsm/kg water. Hydrolyzed peptide formulations, and whey and soya protein hydrolysates can be used in disaccharides or whole protein intolerance [33,47].

### 4.5. Pharmaconutrients

The supplementation of vitamin C, Zinc and selenium arginine, glutamine, and omega 3 fatty acids were not found to significantly impact secondary infections, duration of mechanical ventilation, length of stay, or mortality rates. However, zinc and vitamin D should be administered whenever a deficiency is documented [48].

### 4.6. Electrolytes

Patients with CHD usually require pharmacological intervention in order to properly manage renal function, hypertension, and cardiac failure. Diuretics modify the electrolytes profile in these patients and, therefore, careful monitoring is required. Other than that, the dosage recommendations follow the aforementioned guidelines for age and weight (Table 1).

In the United Kingdom, a study employing Pediatric Dietitians from all the pediatric cardiology surgery centers and using a Delphi process for consensus-based nutritional guidelines, developed three possible nutrition plans, depending on the nutrition risk, infant’s growth, and the infant’s digestive tolerance. Plan A (lower nutritional risk) allows for normal energy and protein requirements, similar to those of a healthy child, and a non-restrictive fluid intake; Plan B involves 10% extra energy requirements and 30–50% increase in protein intake; and Plan C (high nutritional risk) allows up to 20% extra energy requirements and a 50% increase in protein intake. Starting at 17–26 weeks of life, all plans involve adding complementary food, based on protein rich meals, such as ½ − 1 teaspoon of a nut butter or finely ground nuts (plan B and C) and vitamin D supplementation [48,49]; see Table 1**.**

Depending on the CHD, each plan is more likely to be followed. Patients with small septal defects, total anomalous pulmonary drainage, or coarctation of the aorta are more likely to benefit from plan A, while those with pulmonary or tricuspid atresia, prostaglandins-dependent lesion, tetralogy of Fallot, severe septal defects, HLHS, Ebstein’s anomaly, or double outlet right ventricle are considered candidates for plan C.

## 5. Discussion

Most guidelines and protocols for the proper nutrition of newborns with congenital heart defects are based on observational or retrospective studies, which have the limitation of non-standard criteria for assessing the needs of each case; therefore, the risk of initiating inappropriate nutrition due to overestimation or underestimation of a CHD is posed.

Newborns with congenital heart abnormalities have difficulty initiating and sustaining an efficient enteral diet, both due to a waste of energy and to neurological, motor, gastrointestinal, endocrine, and renal developmental problems that occur in association with complex cardiac malformations.

Fluid requirements are particularly difficult to assess, especially in cases where vasoactive and diuretic agents are required. It is necessary to maintain a balance between the minimum enteral diet, which has proven its benefits in preventing gastrointestinal complications both pre and postoperatively, and parenteral nutrition, which must consider the daily fluid requirement and electrolyte balance, while also avoiding fluid overload in the context of a malfunctioning heart.

Parenteral nutrition in newborns and infants can prove to be a major source of oxidants [50]; several studies have suggested that higher arginine and cysteine intake, along with decreased iron and copper concentrations, might help decrease the oxidant load. The range of products used for parenteral nutrition must be strictly controlled in terms of storage conditions and period, exposure to the sun and oxygen, and the temperature of administration. The consumables used can also influence the chemical interactions, leading to the accumulation of oxidants and hepatotoxic and carcinogenic molecules in neonates. For instance, Loff et al. reported a spike in DEHP concentration in lipid solutions from 0.06 µg/mL to 2 µg/mL when using a venous catheter made from PVC–DEHP [51].

Macronutrients, vitamins, and electrolytes administration require great care and a thorough understanding of the general rules of neonate physiology and the pathogenesis of cardiac defects.

## 6. Conclusions

Adequate feeding protocols tailored to meet the requirements of patients with congenital heart defects help improve short- and long-term outcomes. Nutritional and metabolic changes are age-dependent, and because this is a high-risk population, guiding enteral and/or parenteral feeding is difficult. Sources of increased metabolic demand in CHD include increased REE, higher cardiac workload, pulmonary hypertension, and increased catecholamine secretion. Infants and children that receive diuretic therapy experience losses of electrolytes and minerals. Hyponatremia, hypochloremia, and metabolic alkalosis are common disturbances that lead to anorexia, poor weight gain, and impaired wound healing. The available scientific literature and guidelines, if applied to the letter, may help improve nutritional status and outcomes in patients with CHD. The current best practices both in Europe and America have been summarized in this article, albeit missing information regarding the management of complications that may arise in conjuncture with heart defects pre and postoperatively.

## Figures and Tables

**Table 1 nutrients-14-01671-t001:** Electrolytes requirements according to ESPGHAN.

	Infants under 5 kg (Values in mmol/kg/Day)				Infants 5–10 kg (mmol/kg/Day)	Children >10 kg (mmol/kg/Day)
	Day 1	Day 2–3	Day 4–7	Past Day 7		
Sodium	0–2	1–3	2–3	2–3	2–3	1–3
Potassium	0–3	2–3	2–3	1.5–3	1–3	1–3
Calcium	0.8–1.5	0.8–1.5	0.8–1.5		0.5	0.25–0.4
Chloride	0–3	2–5	2–5	2–3	2–4	2–4

**Table 2 nutrients-14-01671-t002:** Optimal nutrition parameters for term ill newborns and preterm neonates with CHDs by means of enteral and parenteral feeding.

	Enteral Feeding	Parenteral Feeding
**Term ill newborn**	Concentrated formula or breast milk	IV fluid
	Start 40–60 kcal/kg/day increase to 90−120 kcal/kg/day	60–70 mL/kg on day 1 increases to 100–120 mL/kg by day 2 or 3
	Carbohydrates 9–14 g/kg/day (40–50% of total calorie intake)	4:2:1 rule
	Proteins 1.8–2.2 g/kg/day (7-16%)	4 mL/kg/h for the first 10 kg weight 2 mL/kg/h for the next 10 kg 1 mL/kg/h past 20 kg
	Lipids 4–6 g/kg/day (34-35%)	
**Preterm**	Gavage feeding (nasal/oral)	IV fluid—70–80 mL/kg on day 1 and slowly advance to 150 mL/kg/day
	iBF (10–20 min infusion every 2–3 h)	Must contain sodium, but avoid hypotonic solutions
	Slow infusion intermittent feeding (30–120 min every 2–3 h)	Electrolytes (Table 1)
	Continuous infusion over 24 h	Proteins—10–15% of the total calorie intake (1 g protein = 4 kcal)
	Semicontinuous feeding (every 15 min throughout the day with ¼ of the hourly volume)	Lipids—30–35% (1 g lipids = 9kcal)
		Carbohydrates—60–65% (1 g glucose = 4 kcal)
**Special Considerations**	Start within 24 h from admission as long as no gastrointestinal anomalies, vomiting, diarrhea, NEC or lactic acidosis are present	Glucose—2.5 mg/kg/min (3.6 g/kg/day) in the acute phase −5.0 mg/kg/min (7.2 g/kg/day) in the recovery phase 10% glucose solutions are preferred
	Formulas rich in protein and energy, but the osmotic load should not exceed 450 mOsm/kg water	Proteins—1.5 g/kg/day for infants and 0.8 g/kg/day for children -in critical CHD—2–3 g/kg/day for ages 0−2 years, 1.5–2 for ages 2–13 years, 1.5 for 13–18 years old
		Lipids—0.5 mg/kg/day intralipid is enough to prevent lipid deficiency -should not exceed 3 g/kg/day
		Pharmaconutrients—Zinc and vitamin D should be adm. whenever a deficiency is documented
		Electrolytes (Table 1)

## Data Availability

All data is available in the present paper.

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
