# Peer review of "Optimal Nutrition Parameters for Neonates and Infants with Congenital Heart Disease"

_nutrients, 2022, doi:10.3390/nu14081671_

Round 1

Reviewer 1 Report

In this review, Luca et al. described the nutritional considerations for neonates and infants with congenital heart disease (CHD). The manuscript can be improved by incorporating following suggestions.

The introduction section should describe the importance of adequate nutrition in children with CHD, obstacles and complications faced by children with CHD across their different phases of care, and the aim of the review.

A significant portion of this review includes general principles of feeding which may not be relevant to the review topic. Authors may want to focus solely on patients with CHD and describe in detail about the importance of nutrition, determinants of nutritional status, caloric targets, nutrition in different phases of cardiac care, and in children with complications secondary to CHD.

Author Response

Dear reviewer 1,

In this review, Luca et al. described the nutritional considerations for neonates and infants with congenital heart disease (CHD). The manuscript can be improved by incorporating following suggestions.

The introduction section should describe the importance of adequate nutrition in children with CHD, obstacles and complications faced by children with CHD across their different phases of care, and the aim of the review.

  • Thank you for your suggestion. We have added a clear sentence pointing out the purpose of the paper, as well as possible complications in the case of children with CHD and how nutrition places a significant role (Lines 46-79).

A significant portion of this review includes general principles of feeding which may not be relevant to the review topic. Authors may want to focus solely on patients with CHD and describe in detail about the importance of nutrition, determinants of nutritional status, caloric targets, nutrition in different phases of cardiac care, and in children with complications secondary to CHD.

  • Thank you for your observation, however, we would like to point out that we in fact described the principles of feeding in chd patients. They are written as “general considerations for enteral/parenteral feeding” but it is in children with heart conditions, general principles for them with a few comparisons to healthy children. We apologize if our subheadings caused any sort of confusing. Our aim was to present general considerations for enteral feeding in chd patients, general considerations for parenteral feeding in chd patients and special considerations for feeding in chd patients. Therefore, we modified the titles.

Thank you for your insightful remarks.

Reviewer 2 Report

Dear Authors,

I have read and mostly appreciated your review article titled Optimal nutrition parameters for neonates and infants with congenital heart disease, a praiseworthy scientific contribution aimed at shedding a light on how prominent a role nutritional management plays in terms of improving short and long-term prognosis for infants with congenital heart disease. Congenital heart disease (CHD) is estimated to affect 1% of newborns yearly, and it is the most commonly occurring major congenital defect. Infants with hemodynamically significant CHD are known to have higher rates of malnutrition and growth failure vs healthy infants.
The article's chief strength resides in its relevance and insightfulness, in addition to its competently enunciated core message. The table outlining enteral vs parenteral approaches via optimal nutritional standards is valuable in its conveying noteworthy information with direct clarity. I would recommend to more comprehensively describe one of the main determinants/contributing factors of CHD children malnutrition:  pulmonary hypertension, more rife in cyanotic heart disease. Hypoxia-related metabolic acidosis is thought to be the leading cause of pulmonary hypertension-induced malnutrition in children with cyanotic heart disease, in addition to chronic hypoxia, anorexia and malabsorption and inadequate processing of nutrients at the cellular level. The authors have not delved into such connections and dynamics deeply enough.
In addition, coherence and organization are serious issues with this manuscript. The authors should tweak the overall structure of the article in order to make it more coherent: there is very little mention of any objective at all. Reviews ought to include a descriptive overview of their fundamental objectives as well as a rather detailed profile of their methodological pathways. Both are all but missing here. Besides, why are the sources cited in brackets with authors'names and matching endnotes in alphabetical order? That is absolutely wrong according to MDPI standards. Also, the endnotes are not at all compliant with MDPI style. I am afraid the authors may have forgotten to read the instructions for authors. Please fix those issues and make your manuscript better organized. It may be worthy of publication by virtue of its relevance and overall clinical value.
Albeit well-written overall, the authors should have the manuscript proof-read by a native speaker of English, in order to fix some occasional faulty grammar and flawed vocabulary choices.

Looking forward to reviewing a substantially improved version.

Author Response

Dear reviewer 2,

I have read and mostly appreciated your review article titled Optimal nutrition parameters for neonates and infants with congenital heart disease, a praiseworthy scientific contribution aimed at shedding a light on how prominent a role nutritional management plays in terms of improving short and long-term prognosis for infants with congenital heart disease. Congenital heart disease (CHD) is estimated to affect 1% of newborns yearly, and it is the most commonly occurring major congenital defect. Infants with hemodynamically significant CHD are known to have higher rates of malnutrition and growth failure vs healthy infants.The article's chief strength resides in its relevance and insightfulness, in addition to its competently enunciated core message. The table outlining enteral vs parenteral approaches via optimal nutritional standards is valuable in its conveying noteworthy information with direct clarity.

  • Thank you for your kind words and we are pleased to know you find our paper insightful and helpful.

I would recommend to more comprehensively describe one of the main determinants/contributing factors of CHD children malnutrition:  pulmonary hypertension, more rife in cyanotic heart disease. Hypoxia-related metabolic acidosis is thought to be the leading cause of pulmonary hypertension-induced malnutrition in children with cyanotic heart disease, in addition to chronic hypoxia, anorexia and malabsorption and inadequate processing of nutrients at the cellular level. The authors have not delved into such connections and dynamics deeply enough. In addition, coherence and organization are serious issues with this manuscript. The authors should tweak the overall structure of the article in order to make it more coherent: there is very little mention of any objective at all. Reviews ought to include a descriptive overview of their fundamental objectives as well as a rather detailed profile of their methodological pathways. Both are all but missing here.

  • We appreciate the suggestion but our paper’s purpose was to highlights the specific metabolic changes in patients with CHD and to summarize studies regarding appropriate enteral and parenteral nutrition (as stated in lines 46-47). However, we have added some extra paragraphs describing main determinants of CHD malnutrition and other possible connections (Lines 49-78). We hope that they improve the paper in the direction that you have suggested.

Besides, why are the sources cited in brackets with authors'names and matching endnotes in alphabetical order? That is absolutely wrong according to MDPI standards. Also, the endnotes are not at all compliant with MDPI style. I am afraid the authors may have forgotten to read the instructions for authors. Please fix those issues and make your manuscript better organized. It may be worthy of publication by virtue of its relevance and overall clinical value.
Albeit well-written overall, the authors should have the manuscript proof-read by a native speaker of English, in order to fix some occasional faulty grammar and flawed vocabulary choices. Looking forward to reviewing a substantially improved version.

  • I apologize for the references. Must have been an error or mistake when submitting the manuscript, we have corrected the issue and now the references as well as citations are in the proper style and order. And we also went over the text once again and modified where we saw needed.

Thank you for all of your comments and praise words and we hope you will find our manuscript to be improved.

Round 2

Reviewer 1 Report

Authors addressed all my concerns. The manuscript is ready for publication in my opinion.

Reviewer 2 Report

Dear Authors,

You have further improved and broadened the scope of your article by properly addressing the reviewer's reamarks.

The new part on nutritional status determination is a valuable addition that makes the manuscript even more comprehensive.

All the best with your future endeavors.